# Association of Physical Performance with Mental and Physical Health-Related Quality of Life and Low Back Pain-Related Disabilities among Older Adults with Severe Obesity

**DOI:** 10.3390/jcm13185614

**Published:** 2024-09-22

**Authors:** Munkh-Erdene Bayartai, Gabriella Tringali, Roberta De Micheli, Ana Lúcia Danielewicz, Alessandro Sartorio

**Affiliations:** 1Department of Physical and Occupational Therapy, School of Nursing, Mongolian National University of Medical Sciences, Ulaanbaatar 14210, Mongolia; 2Istituto Auxologico Italiano, Istituto di Ricovero e Cura a Carattere Scientifico (IRCCS), Experimental Laboratory for Auxo-Endocrinological Research, 28824 Piancavallo-Verbania, Italy; g.tringali@auxologico.it (G.T.); r.demicheli@auxologico.it (R.D.M.); sartorio@auxologico.it (A.S.); 3Department of Health Sciences, Postgraduate Program in Rehabilitation Sciences, Federal University of Santa Catarina, Araranguá 88906-072, Santa Catarina, Brazil; ana.lucia.d@ufsc.br

**Keywords:** low back pain, disability, obesity, health-related quality of life

## Abstract

**Background:** Low back pain is one of the most prevalent musculoskeletal problems and continues to be the leading cause of disabilities worldwide. The aim of this study was to cross-sectionally investigate the association of physical performance with mental and physical health-related quality of life and low back pain-related disabilities among older adults with severe obesity. **Methods:** A total of 96 hospitalized older adults with severe obesity (45 males, 51 females, age: 69.7 ± 5.4 years; BMI: 43.7 ± 5.7 kg/m^2^) were recruited into the study. Physical performance, health-related quality of life, and low back pain-related disability were measured through physical performance tests, the 12-item short-form survey (SF-12), and the Oswestry disability index, respectively. **Results:** LBP-related disabilities, as well as physical health-related quality of life, were associated with all the physiological parameters measured by physical performance tests, including muscular strength, aerobic capacity, balance, and lower body flexibility (*p* < 0.05). In contrast, mental health-related quality of life was associated with fewer physiological parameters, such as primarily muscular strength (*p* < 0.05). **Conclusions:** These findings could provide important insights for developing rehabilitation strategies designed to improve LBP-related disabilities, as well as the physical and mental health-related quality of life, in older adults with severe obesity.

## 1. Introduction

It is well established that the consequences of musculoskeletal problems frequently lead to disabilities, which are multifactorial and complex, exacerbated by the interaction of an individual’s health condition with environmental and personal factors [1,2]. Worldwide, low back pain (LBP) is one of the most common musculoskeletal disorders that contribute to disabilities [3,4]. In 2020, the global age-standardized rate of years lived with disability was 832 per 100,000 individuals, with 619 million people suffering from LBP [4]. This number is expected to rise to 843 million by 2050 [3,4].

Chronic LBP is associated with lower health-related quality of life, physical impairments, and disabilities [5,6]. A cross-sectional study involving 95 women identified decreased lower extremity strength as a critical predictor of reduced physical performance [6]. Prospective studies also demonstrated that lower hand grip strength was associated with more significant disabilities in older populations [7,8]. Additionally, as people age, they typically experience sarcopenia characterized by diminished muscle mass and strength along with a reduction in functional capacity [9,10,11]. Although different physiological parameters, such as muscular strength, endurance, flexibility, and stability/balance, can be measured by different physical performance tests, studies with other physical performance measures as predictors of disabilities and low quality of life in older adults with obesity and chronic LBP are lacking.

The prevalence of obesity tends to increase with age, particularly from the age of 65 until 75 [12]. Obesity is also associated with various health conditions, including LBP, and is linked to a variety of impairments in mobility and self-care, leading to disabilities [13,14,15]. A systematic review involving both cross-sectional and longitudinal studies revealed that individuals with obesity had a greater risk of limitations to activities of daily living than those with normal weight [13]. Another review with cross-sectional and longitudinal studies of older adults with obesity reported that, despite the challenges in interpreting different physical performance tests, a consistent finding was that severe obesity (body mass index ≥ 35 kg/m^2^) hindered abilities such as walking, stair climbing, and chair rising [14]. The authors of this review also suggested that body mass index (BMI) and waist circumference are becoming the most reliable indicators of the onset or progression of mobility disabilities [14]. Although impairments in physical performance in relation to disabilities have extensively been studied, studies examining the association between LBP-related disabilities, health-related quality of life, and physical performance in older individuals with severe obesity are lacking.

Identifying reduced physical performance pertinent to obesity and low back pain could provide valuable insights into rehabilitation strategies aimed at addressing disabilities and improving quality of life. Therefore, the present study aims to explore the association of different physical performance tests, including muscular strength, endurance, flexibility, and stability/balance with mental and physical health-related quality of life and LBP-related disabilities among older adults with severe obesity.

## 2. Materials and Methods

The present study employed a cross-sectional design, following the Strengthening the Reporting of Observational Studies in Epidemiology (STROBE) guidelines (Appendix A) [16].

### 2.1. Participants

This study was conducted with Italian older adults, of both sexes, with severe obesity, hospitalized between April 2023 and November 2023 at the Division of Pneumological Rehabilitation and the Division of Rehabilitative Medicine, Istituto Auxologico Italiano, IRCCS, Piancavallo-Verbania, Italy (Figure 1). These individuals participated in a 3-week in-hospital multidisciplinary body weight reduction program, entailing an energy-restricted diet, nutritional education, psychological counseling, and physical rehabilitation (moderate aerobic activity), as previously described [17]. A post hoc power analysis was conducted to assess the adequacy of the sample size using the G*Power 3.1 tool [18]. The analysis revealed that with a sample size of 96 and an effect size (Cohen’s f) of 0.36, along with an alpha level of 0.05, the power was 0.88, indicating that the sample size was adequate for the analysis. The effect size was derived from the association between physical performance and LBP-related disability in the present study.

The inclusion criteria were age 60 and over and a BMI of 35 kg/m^2^ or more. Participants were excluded from the study if they had prosthetics, were completely unable to walk, or had evident cardiopulmonary or metabolic diseases hindering physical exertion. The study was approved by the Ethics Committee of Istituto Auxologico Italiano, Milan, Italy (registration code: 2023_03_21_07; date of approval: 21 March 2023; research project code: 01C313), and was conducted in compliance with the Helsinki Declaration of 1975, as amended in 2008. The purpose and objective of the study were explained to each participant, and written informed consent was obtained from all eligible participants.

The present study recruited 96 participants, who were categorized into three groups: those with minimal disability related to low back pain (LBP), those with moderate LBP-related disability, and those with no lower back pain.

### 2.2. Measurements

#### 2.2.1. Assessment of Morbidities

The number of following self-related diseases was evaluated: low back pain, arthritis, cancer, diabetes, hypertension, bronchitis or asthma, sleep apnea, cardiovascular disease, kidney failure, brain stroke, osteoporosis, labyrinthitis, and urinary incontinence.

#### 2.2.2. Anthropometric Parameters’ Assessments

Health professionals were trained for measurements of anthropometric parameters. Body mass index (BMI) was determined by dividing body weight by the square of height, expressed in kg/m^2^. Waist circumference (cm) was measured while standing, halfway between the lowest rib and the iliac crest.

#### 2.2.3. Assessment of Physical Performance Tests

All the assessments were performed in the first three days of hospitalization. The Short Physical Performance Battery (SPPB) and the physical performance test (PPT) were used to determine physical performance. The SPPB was developed by Guralnik et al. in 1994 to evaluate balance and physical function, with a particular focus on lower extremity function [19]. The SPPB consists of three main components: standing balance (standing with feet together in three progressively challenging positions: side-by-side, semi-tandem, and tandem), gait speed (measured over a four-meter course at a normal pace), and the ability to rise from a chair, designed to evaluate lower extremity strength (time to stand up five times from a chair with arms crossed over the chest). Each of the three tasks is scored from 0 (unable to perform the task) to 4 (optimal performance), and these scores are then added together to produce a final SPPB score, which ranges from 0 to 12. Lower total scores indicate reduced physical functioning.

The PPT, developed by Rueben and Siu in 1990, is a frequently used tool to measure the overall functional ability of older individuals [20]. The short version of the PPT, consisting of 7 functional tasks, was used in the current study. The functional tasks on the short version include writing a sentence, simulated eating, lifting a book and putting it on a shelf, donning and doffing a jacket, picking up a penny from the floor, turning 360 degrees while standing, and walking 15.2 m [21]. Each of the seven tasks is scored from 0 to 4 points, which are then summed to calculate the overall PPT score. The available evidence supports the PPT’s inter-rater reliability, concurrent validity, and predictive validity for measuring physical function in elderly individuals [20,22,23,24].

#### 2.2.4. Assessment of Hand Grip Strength (HGS)

The hand grip strength (HGS) was measured with a hand dynamometer (Lafayette Instrument, Inc., Lafayette, LA, USA) in a sitting position, with the shoulder and wrist in a neutral position and the elbow at 90 degrees of flexion [25]. Three measurements were performed with the dominant hand, and the average value was used in the analyses.

#### 2.2.5. Assessment of Aerobic Capacity (6 Min Walk Test)

The American Thoracic Society developed the six-minute walking test (6MWT) to evaluate aerobic capacity and endurance [26]. The 6MWT was performed for all the participants. The individuals were instructed to walk as fast as they could along an even, undisturbed 30 m hospital corridor marked every 5 m; the operator used a lap counter system, and the complete distance walked during 6 min was measured using a tape measure from the nearest marker with colored tape on the floor [27,28]. Encouragement was given every minute during the test until subject exhaustion using only standardized phrases as specified in the “ATS Statement: Guidelines for the Six-minute Walk Test” [28]. Chest pain, severe dyspnea, physical exhaustion, muscle cramps, sudden gait instability, or other signs of severe distress were additional criteria for stopping the test [28]. The individual’s pulse, respiratory rate, blood pressure, and perceived fatigue as assessed on Borg’s scale [28] were measured before the test and at test completion. The distance covered in 6 min by each participant was used as a variable for the analysis. They were allowed to slow down, stop, and rest during the test if they felt fatigued or had shortness of breath.

#### 2.2.6. Assessment of Lower Limb Power, Strength, and Function

The stair climb test was used to evaluate an individual’s performance in climbing and descending stairs, aimed to assess the power, strength, and function of lower limbs. Participants were asked to climb a 10-step flight of stairs as quickly as they felt safe and comfortable, turn around, and come back down. Time spent during the performance between the first step to climbing the stairs and the last step, where the second foot touches the landing, was recorded using a stopwatch [29,30].

#### 2.2.7. Assessment of Senior Fitness Test (STF)

A senior fitness test (STF) comprising six functional tests was used to evaluate the physical function of all older participants. Each functional test in the SFT is scored using distinct scales [31]. The six functional tests cover the chair stand test, the biceps curl test, the 2 min walk test, the chair sit and reach test, the back scratch test, and the 8-foot up and go test aimed at evaluating lower body strength, upper body strength, aerobic endurance, lower body flexibility, upper body flexibility, and agility/dynamic balance, respectively. Previous studies demonstrated that the SFT is reliable with an ICC (intraclass correlation coefficient) between 0.8 and 0.9 [31,32,33,34]. The validity of the functional tests of SFT for measuring body strength, aerobic endurance, and body flexibility by comparing with gold standards for these physiological attributes was found to be moderate, ranging from r = 0.71 to r = 0.84 [31,32,33,34].

#### 2.2.8. Assessment of Quality of Life

The 12-item short-form health survey (SF-12) assessed the participants’ quality of life, containing 12 questions exploring various physical and mental health aspects. Participant’s responses to the SF-12 questionnaire were carefully evaluated, and the physical and mental component scores were calculated according to the SF-12 scoring algorithm [35]. A score of 50 on the SF-12 indicates average health. Therefore, scores above 50 represent a superior health-related quality of life, whereas scores below 50 imply an inferior health status relative to the average. A cross-sectional study involving 1343 individuals aged between 60 and 85 demonstrated satisfactory internal consistency reliability and discriminant validity of the SF-12, with Cronbach’s α value of 0.91 and Spearman’s ρ > 0.4, respectively [36]. Another study involving 1587 individuals with either a combination of physical and behavioral conditions or severe mental illness examining the reliability of the SF-12 found ICCs of 0.61 and 0.57 for the Physical Composite Scale and the Mental Health Composite Scale, respectively [37].

#### 2.2.9. Assessment of LBP-Related Disability

The Oswestry disability index (ODI), a reliable and robust tool for measuring patient-reported disability, was used to evaluate participants’ disability levels [38]. The ODI consists of 10 items related to functional activities of daily living, such as pain intensity, personal care, lifting, walking, sitting, standing, sex life, social life, and traveling. Patients select one of six statements for each item, with scores ranging from 0 to 5, reflecting their ability to perform these activities while considering their pain levels. The overall score for all items is calculated by dividing the sum of the item scores by the maximum possible score and then multiplying by 100 to yield a percentage of disability. Consequently, the ODI total score ranges from 0 (indicating no disability) to 100 (indicating maximum disability). In this study, only 7 out of the 10 ODI items were used to assess disability, as the last 3 items were irrelevant for hospitalized patients. Participants were then classified into three distinct groups based on their ODI scores: minimal disability (0–20% ODI), moderate disability (21–40% ODI), and severe disability (41–80% ODI).

### 2.3. Statistical Analysis

Descriptive statistics and inferential analyses were conducted using R version 3.6.0 [39]. For the descriptive statistics, mean values and standard deviations (SDs) for the participant characteristics of age, sex, weight, height, BMI, waist circumference, the ODI disability score, and physical and mental component scores of the SF-12 were determined. The Shapiro–Wilk test was applied to check for data normality (Appendix A). Analysis of variance (ANOVA) for normally distributed data and a Kruskal–Wallis test for non-normally distributed parameters were used to compare participants’ characteristics between individuals with and without chronic LBP. The chronic LBP group was further categorized into two subgroups according to the severity of disability related to LBP. The chi-square test was used for categorical variables. A two-way ANOVA, adjusting for sex and morbidities, was used to determine statistically significant physical performance, fitness, HGS, and quality of life differences between the three groups. Pairwise post hoc tests were performed following the ANOVA tests using the software package “emmeans v 1.6.3” [40]. Linear regression was used to determine the association of physical performance with mental and physical health-related quality of life and LBP-related disability. The Bonferroni adjustment was applied to all multiple comparisons to ensure that the overall error rate remained at 0.05. The strengths of associations were compared using standardized beta coefficients. Standardized beta coefficients are estimated in standard deviation units, allowing us to compare the strength of associations of different independent variables with the dependent variable. Standardized coefficients were calculated by dividing the multiplication between the raw regression coefficient and the independent variable’s standard deviation by the dependent variable’s standard deviation. *p* values less than 0.05 were considered to be statistically significant.

## 3. Results

A total of 96 older adults with severe obesity (45 males, 51 females, age: 69.7 ± 5.4 years; BMI: 43.7 ± 5.7 kg/m^2^) were analyzed. In total, 71 out of 96 participants reported that they experienced chronic low back pain (pain persisting for more than three months), whereas 25 had no previous history of low back pain (Table 1). Participants’ characteristics by LPB-related disabilities are provided in Table 1. No mean age, BMI, or waist circumference differences were observed between the minimal, moderate, or no LBP-related disability groups. In contrast, the mean height, sex prevalence, and number of morbidities were different across the three groups (Table 1). Older adults with LBP had more significant disabilities compared to those without LBP. Furthermore, older adults with moderate LPB-related disabilities had lower mental and physical health-related quality of life than those with minimal or without LBP-related disabilities.

### 3.1. Physical Performance in Older Adults with and without Chronic LBP

Participants with LBP and moderate disabilities had lower scores in physical performance tests measuring muscular strength, aerobic capacity, balance, and flexibility than those with minimal disabilities as well as those without LBP disabilities (Table 2). There was an exception for HGS and chair sit and reach (SFT4) tests assessing hand strength and lower body flexibility, respectively, which were greater only in participants with minimal LBP-related disabilities than those with moderate LBP-related disabilities. No statistically significant differences were observed in scores of physical performance tests between individuals with minimal LBP-related disabilities and those without LBP-related disabilities. Upper body flexibility, as assessed by the back scratch test (SFT5), was the only physiological parameter that showed no significant differences across the three groups (Table 2).

### 3.2. Associations of Physical Performance Tests with Disabilities and Quality of Life among Older Adults with Chronic LBP

Physical performance, measured by SPPB, was the strongest determining factor for LBP-related disability and physical health-related quality of life. In contrast, body strength (upper and lower equally) was the key predictor of mental health-related quality of life (Table 3). All parameters measured by the physical performance tests were associated with LBP-related disabilities as well as physical health-related quality of life to a certain extent. Fewer physiological parameters were related to mental health-related quality of life compared to those with LBP-related disabilities and physical health-related quality of life. For example, after the Bonferroni adjustment of *p*-values, only the association of upper and lower extremity strength with mental health-related quality of life remained significant (Table 3).

## 4. Discussion

The purpose of the present study was to investigate the association of physical performance tests with mental and physical health-related quality of life and LBP-related disabilities among older adults with severe obesity. The main findings were that LBP-related disabilities, as well as physical health-related quality of life, were associated with all the physiological parameters measured by physical performance tests, including muscular strength, aerobic capacity, and balance. In contrast, mental health-related quality of life was associated with fewer physiological parameters, primarily muscular strength.

All physiological parameters measured by physical performance tests were associated with LBP-related disability, with aerobic capacity being the strongest predictor, followed by muscular strength among older adults with chronic LBP and severe obesity. Previous studies found that physiological parameters such as lower hand grip strength and slower gait speed were associated with disability related to hospitalizations and difficulties with activities of daily living among older populations [7,8,41]. Older adults with moderate disabilities demonstrated significant impairments in physical performance compared to those with minimal disabilities and those without LBP, while physical performances were not different between the latter two groups. This result implies that the transition from minimal to moderate LBP-related disabilities was associated with a significant decline in physical performance in older adults with severe obesity.

The present study also revealed that lower body flexibility was associated with greater LBP-related disability. In contrast, upper body flexibility showed no association with the severity of LBP-related disability. The American College of Sports Medicine recommends engaging in various types of exercise, including aerobic activities, strength training, and flexibility exercises, to maintain or improve flexibility and preserve balance, thereby preventing falls and reducing morbidity in older adults [42]. Reduced lumbar flexibility is also associated with LBP [43]. A systematic review of prospective cohort studies reported that restricted lateral bending motion and decreased hamstring flexibility, as well as reduced lumbar lordosis, were predictors of the development of LBP [44]. Nevertheless, the relationship between flexibility and LBP-related disability has been understudied in research, even among general populations to date. Therefore, physiological parameters associated with LBP-related disability, particularly those understudied, such as body flexibility, must be further examined in longitudinal studies in elders with chronic LBP and obesity. This could provide valuable insights for developing rehabilitation strategies aimed at improving LBP-related disability among older individuals suffering from chronic LBP and obesity.

Both the PPT and SPPB were also associated with LBP-related disabilities, with the SPPB showing a stronger association. Aerobic capacity, assessed by the 6 Min walking test, was the second strongest predictor of LBP-related disability. This could explain the strong relationship between the SPPB and LBP-related disability, as aerobic capacity is one of the three main components assessed by the SPPB. Scores of all the physical performance tests designed to measure upper and lower body strength as well as hand strength were associated with LBP-related disabilities, highlighting the significance of different muscular strength assessments for these clinical conditions. A previous study examining the accuracy of handgrip strength and gait speed, separately and combined, in identifying functional disabilities in older Chinese adults demonstrated that the gait speed test was better than the handgrip strength test in detecting this outcome [41]. A previous comparative study of different physical performance tests, including the SPPB, 4 m walk test, 6 Min walk test, and handgrip strength test, demonstrated that all these physical performance measures could predict disabilities in activities of daily living in older people [45]. These results from the earlier studies, although not generalizable to older individuals with chronic LBP and severe obesity, were consistent with the findings of the present study. Balance was associated with LBP-related disabilities, but their association was substantially weaker than that of muscular strength and aerobic capacity.

Physical performance, measured by the SPPB, followed by aerobic capacity and muscular strength, were the primary predictors of physical health-related quality of life. The key predictors of physical health-related quality of life were also the strongest predictors of LBP-related disability, indicating a strong link between physical health-related quality of life and LBP-related disability. Although studies examining the relationship between physical performance and health-related quality of life, specifically among older adults with chronic LBP and severe obesity, are lacking, this association has been investigated in other populations. For instance, as measured by the SPPB, greater physical function was associated with a higher level of physical health-related quality of life in older adults who have or are at risk of mobility disability after being discharged from the hospital [46]. A cross-sectional study involving older adults aged between 60 and 80 revealed that physical performance, measured by standing balance, walking speed, and grip strength, was positively correlated with the physical domain of quality of life [47], consistent with findings from the present study.

Fewer physiological parameters, primarily muscular strength, were associated with mental health-related quality of life compared to those linked to LBP-related disability and physical health-related quality of life. The strongest predictor of mental health-related quality of life was upper and lower body strength, followed by grip strength. However, after the Bonferroni adjustment of *p*-values for multiple comparisons, only the association of upper and lower extremity strength with mental health-related quality of life remained significant, emphasizing the importance of muscular strength in predicting mental health-related quality of life. Anxiety and depression, as aspects of quality of life, have previously been investigated in relation to physical performance in community-dwelling adults aged over 70. The research found that higher levels of depression and anxiety were associated with poorer performance on the stair climb and chair stand tests [48]. A previous systematic review also indicated that muscular strength was inversely associated with depression symptoms [49], supporting the findings of the present study. The PPT and SPPB were also able to predict mental health-related quality of life, but their association with mental health-related quality of life was no longer significant after adjusting the significance level for multiple comparisons in the present study. These results suggest that further research, particularly prospective studies, is needed to better understand the relationship between physical performance and mental health-related quality of life.

The findings from the present study could provide valuable insights for developing rehabilitation strategies aimed at promoting mental health among older adults with chronic LBP and severe obesity. This is especially relevant given the scarcity of comparative studies on different physical performance measures as predictors of mental health-related quality of life in this population. However, this hypothesis should be confirmed through a randomized controlled trial.

We acknowledge that the present preliminary study has several limitations. We employed a cross-sectional design, which precludes any causal interpretations of the observed associations of physical performance with mental and physical health-related quality of life and LBP-related disability. Participants of the present study were limited to those with minimal to moderate LBP-related disability and severe obesity. Therefore, our findings could not be generalized to all older individuals with severe LBP-related disabilities or older populations without severe obesity. Additionally, although the correction of multiple comparisons was performed, the relatively small sample size could increase the risk of a Type I error. Further additional studies are requested to give definitive answers to these aspects not addressed by this preliminary study.

## 5. Conclusions

LBP-related disabilities and physical health-related quality of life were associated with all the physiological parameters measured by physical performance tests, including muscular strength, aerobic capacity, balance, and flexibility, except for upper body flexibility. In contrast, mental health-related quality of life was associated with fewer physiological parameters, such as primarily muscular strength. These findings could provide valuable insights for developing rehabilitation strategies aimed at improving LBP-related disabilities and physical and mental health-related quality of life among older adults with severe obesity.

## Figures and Tables

**Figure 1 jcm-13-05614-f001:**
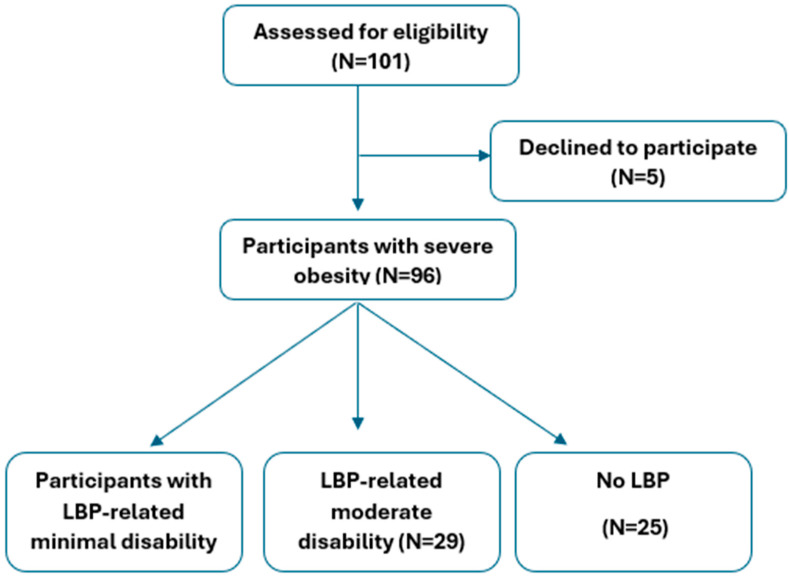
Flowchart of the study participants.

**Table 1 jcm-13-05614-t001:** Participants’ characteristics by the presence of LBP-related disabilities.

Variables	Chronic LBP (n = 71)		No LBP n = 25	*p*-Value
	Minimal LBP-Related Disabilities n = 42	Moderate LBP-Related Disabilities n = 29		
Age (mean; ±SD)	68.9 (4.7)	70.8 (6.2)	69.7 (5.4)	>0.05 ^A^
Sex (female, %)	52.0	17.0	72.0	<0.05 *^,#,C^
Weight (kg) (mean; ±SD)	116.4 (23.1)	112.5 (16.1)	122.1 (23.3)	>0.05 ^K^
Height (cm) (mean; ±SD)	164 (11.1)	158 (9.6)	167 (9.1)	0.02 *^,K^ 0.002 ^#,K^
BMI (kg/m^2^) (mean; ±SD)	42.8 (5.4)	45.1 (4.7)	43.3 (6.9)	>0.05 ^K^
Waist circumference (cm) (mean; ±SD)	129.6 (14.9)	129.9 (8.5)	134.6 (13.8)	>0.05 ^A^
Morbidities (mean; ±SD)	4.3 (2.1)	5.4 (2.0)	3.9 (1.8)	0.01 *^,K^ 0.008 ^#,K^
Disability by ODI index (mean; ±SD)	11.6 (5.9)	27.2 (4.2)	7.9 (7.9)	<0.0001 *^,#,K^ 0.008 *^*^,K^
Quality of life (Physical component score) (mean; ±SD)	34.5 (7.2)	26.6 (4.9)	40.0 (7.6)	<0.0001 *^,A^ 0.004 *^*^,A^ <0.0001 ^#,A^
Quality of life (Mental component score) (mean; ±SD)	50.8 (10.5)	40.2 (11.1)	51.4 (9.6)	0.0004 *^,#,K^

* LBP with minimal disability vs. LBP with moderate disability *p* < 0.05, ^ LBP with minimal disability vs. without LBP *p* < 0.05, # LBP with moderate disability vs. elders without LBP *p* < 0.05, *p*-value—statistical significance computed by using ANOVA ^A^, Kruskal–Wallis test ^K^, and the chi-square test ^C^ for a comparison between the three groups, SD—standard deviation, ODI—Oswestry disability index, LBP—low back pain.

**Table 2 jcm-13-05614-t002:** Physical performance tests by the presence of LBP-related disabilities.

Variables		Chronic LBP (n = 71)		No LBP n = 25 EMM (SE)	*p*-Value
Physical Performance Tests	Measured Physiological Parameters	Minimal LBP-Related Disabilities n = 42 EMM (SE)	Moderate LBP-Related Disabilities n = 29 EMM (SE)		
Hand grip (kg)	Strength (Hand)	27.2 (1.1)	22.0 (1.3)	25.8 (1.4)	0.01 *
6 Min walking test (m)	Aerobic capacity	371 (16.3)	261 (23.9)	349 (23.8)	0.001 * 0.04 ^#^
Stair climbing test (sec)	Strength (Lower body)	9.1 (0.8)	15.4 (1.7)	9.7 (1.0)	0.003 * 0.01 ^#^
Physical performance by SPPB					
Balance (pt)	Balance	3.5 (0.1)	2.5 (0.2)	3.4 (0.2)	0.0005 * 0.01 ^#^
Gait speed (pt)	Aerobic capacity	3.0 (0.1)	1.8 (0.2)	2.6 (0.2)	0.0001 * 0.01 ^#^
Lower extremity strength (pt)	Strength (Lower body)	2.4 (0.1)	1.4 (0.2)	1.9 (0.2)	0.0005 *
Physical performance (total)	Total	9.0 (0.3)	5.7 (0.4)	7.9 (0.4)	<0.0001 * 0.003 ^#^
Physical performance by PPT					
Physical performance (total)	Total	20.6 (0.4)	14.1 (0.9)	20.2 (0.9)	<0.0001 * 0.0001 ^#^
Senior Fitness					
SFT 1 (30 s chair stand)	Strength (Lower body)	10.7 (0.5)	7.1 (0.6)	9.5 (0.7)	0.0001 * 0.03 ^#^
SFT 2 (30 s arm curl)	Strength (Upper body)	15.9 (0.6)	12.0 (0.8)	15.8 (0.8)	0.001 * 0.01 ^#^
SFT 3 (2 min step)	Aerobic capacity	107.8 (7.8)	41.5 (10.2)	98.3 (10.5)	<0.0001 * 0.001 ^#^
SFT 4 (chair sit and reach test)	Flexibility (lower body)	−13.2 (2.1)	−23.1 (2.7)	−15.6 (2.8)	0.01 *
SFT 5 (back scratch test)	Flexibility (upper body)	−23.2 (1.9)	−30.4 (2.9)	−23.5 (2.5)	>0.05
SFT 6 (8-foot up and go test)	Balance (dynamic)	8.9 (0.6)	13.2 (0.8)	9.0 (0.7)	0.0001 * 0.001 ^#^

*p*-value (adjusted for sex and the number of morbidities)—the significance of differences between the groups, EMM—estimated marginal mean, SE—standard error, * LBP with minimal disability vs. LBP with moderate disability *p* < 0.05, # LBP with moderate disability vs. without LBP *p* < 0.05, SFT—senior fitness test, PPT—physical performance test, SPPB—Short Physical Performance Battery, LBP—low back pain.

**Table 3 jcm-13-05614-t003:** Associations of physical performance tests with disability and quality of life in older adults with LBP.

Variables	Disability (ODI)			Quality of Life (SF-12)					
				PCS-12			MCS-12		
	B	B^a^	*p*.adj	B	B^a^	*p*.adj	B	B^a^	*p*.adj
Physical performance by PPT	−0.77	−0.49	<0.0001	0.51	0.41	0.002	0.54	0.27	0.14
Physical performance by SPPB	−1.9	−0.64	<0.0001	1.31	0.54	<0.0001	0.99	0.25	0.23
Balance (pt)	−3.2	−0.41	0.002	2.23	0.36	0.01	1.29	0.13	1.0
Lower extremity strength (pt)	−4.3	−0.54	<0.0001	2.56	0.40	0.003	3.21	0.32	0.04
Upper extremity strength (30 s arm curl test)	−0.83	−0.45	0.0006	0.54	0.37	0.01	0.76	0.32	0.04
Hand grip strength (kg)	−0.43	−0.54	<0.0001	0.26	0.42	0.002	0.23	0.23	0.33
6 Min walking test (m)	−0.05	−0.59	<0.0001	0.03	0.47	0.001	0.01	0.11	1.0

B—raw regression coefficient, B^a^—standardized regression coefficient, *p*.adj—Bonferroni adjusted *p*-value, PPT—physical performance test, SPPB—Short Physical Performance Battery; SF-12—12-item short-form health survey, PCS-12—physical component score of the SF-12; MCS-12—mental component score of the SF-12.

## Data Availability

Raw data will be available on www.zenodo.org after the acceptance of the manuscript upon a reasonable request to the corresponding author.

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
