# Peer review of "Association of Physical Performance with Mental and Physical Health-Related Quality of Life and Low Back Pain-Related Disabilities among Older Adults with Severe Obesity"

_jcm, 2024, doi:10.3390/jcm13185614_

Round 1

Reviewer 1 Report

Comments and Suggestions for Authors

The major limitations of the study are the small sample size and the cross-sectional design that make questionable the reliability of the results! Not only, but also the statistical analysis is arguable, usually regression analysis models do not hold and become problematic, when including high number of variables with small sample size. Moreover in the case of multi-comparison the adjustment of level of significance has not been taken into consideration. It is not clear why authors did not recruited larger sample. 

In this direction because of the cross-sectional design along the entire manuscript any cause-effect conclusion should be removed and authors should limit their statements to simple associations.

I have some concerns related to authorship. Well the study has been conducted in Italy, however the first author is from Mongolia, and his contributions was the formal analysis and writing the manuscript, while in general the fist author is the one who conducted the investigation. Moreover A.L.D. has put under the investigators, but she is in Brazil, where the study has been conducted in Italy.

Authors reported that the study has been conducted according to STROBE guidelines. Authors are requested to submit the suitable checklist. 

Authors reported that the Italian Ministry of Health supported their work. However it is not clear how! Authors are requested to mention how.

The multicolour figure should be removed it is not readable neither. 

Author Response

Reviewer 1

We acknowledge the Reviewer for carefully evaluating our manuscript, which is aimed at improving the quality of our work further.

Q1. The major limitations of the study are the small sample size and the cross-sectional design that make questionable the reliability of the results!

A1. The huge number of assessments scheduled by the study protocol and the peculiarities of the population studied (hospitalized, severely obese, older) hampered the recruitment of a greater number of patients. However, a post hoc power analysis has been conducted to assess the adequacy of the sample size, confirming that the sample size was adequate for the analysis. This lacking information has been added in the materials and methods section. Furthermore, the present study must be intended as a preliminary observation, as we have underlined now in the limitations of the study.

Lines (changes made): 83-88

Q2. Not only, but also the statistical analysis is arguable, usually regression analysis models do not hold and become problematic, when including high number of variables with small sample size. Moreover in the case of multi-comparison the adjustment of level of significance has not been taken into consideration. It is not clear why authors did not recruited larger sample. 

A2. We have taken the reviewer's concerns about the statistical analysis seriously. We have made significant improvements by reducing the number of variables and adjusting the significance level for multiple comparisons as suggested. We have also acknowledged the relatively small sample size used in the present study in the limitation section. Due to the Bonferroni adjustment, the results have been slightly affected, leading to minor modifications in the discussion section as well.

Lines (changes made): 218, 219, 269-271, 273-274, 313-316, 330, 348-350, 356-361, 374-377

Q3. In this direction because of the cross-sectional design along the entire manuscript any cause-effect conclusion should be removed and authors should limit their statements to simple associations.

A3. As recommended, we have limited our statements to simple associations.

Q4. I have some concerns related to authorship. Well the study has been conducted in Italy, however the first author is from Mongolia, and his contributions was the formal analysis and writing the manuscript, while in general the fist author is the one who conducted the investigation. Moreover A.L.D. has put under the investigators, but she is in Brazil, where the study has been conducted in Italy.

A4. All the authors agreed upon the authorships (and their relative position) before submitting the manuscript. The study coordinator (prof. Alessandro Sartorio), director of the Experimental Laboratory for Auxo-endocrinological Research at Istituto Auxologico Italiano, Piancavallo-Varbania (Italy), where the study was conducted, is positioned in the last position.  Prof. Ana Lucia Danielewicz, Federal University of Santa Catarina, Araranguá, Santa Catarina, Brazil, spent a one-year postdoctoral abroad at the Experimental Laboratory for Auxo-endocrinological Research, Istituto Auxologico Italiano, Piancavallo-Verbania (Italy), coordinating the study population recruitment. This lacking information has been added in the affiliation of the authorships and acknowledgments. Prof. Munk-Erdene Bayartai is an external collaborator of the Experimental Laboratory for Auxo-endocrinological Research, Istituto Auxologico Italiano, Piancavallo-Varbania (Italy), as documented by several papers published together in the last years with a comparable approach in the authorships (ref. J Clin Med. 2024 Apr 7;13 (7): 2135; Front Endocrinol (Lausanne) 2023 Sep 20; 14: 1235030; Sci Rep. 2023 Aug 17; 13 (1): 13409; Sci Rep. 2022 Sep 16; 12 (1): 15570; J Clin Med. 2022 Jun 2; 11 (11): 3175). We confirm the correctness of the authorships, which we believe to be appropriate based on the actual roles played by the single Authors in the present project.

Q5. Authors reported that the study has been conducted according to STROBE guidelines. Authors are requested to submit the suitable checklist. 

A5. The checklist has been added.

Q6. Authors reported that the Italian Ministry of Health supported their work. However it is not clear how! Authors are requested to mention how.

A6. The Italian Ministry of Health supports Istituto Auxologico Italiano, a national Scientific Institution for Hospitalization and Care (i.e., IRCSS), for developing Current research (i.e., Ricerca Corrente). As clearly indicated in the affiliations, the Experimental Laboratory for Auxo-endocrinological Research, directed by prof. Alessandro Sartorio, is one of the Experimental Laboratories of this scientific Institution.    

Q7. The multicolour figure should be removed it is not readable neither. 

A7. As requested, the figure has been removed from the manuscript.

Reviewer 2 Report

Comments and Suggestions for Authors

The article addresses the association of topics of significant interest to the journal's readers, such as physical performance, quality of life, disability, and low back pain in obese individuals. The article is well-structured and contains all the sections of an original research paper.

The introduction is well-crafted, although it is somewhat brief, which does not facilitate a clear focus on the problem. The methodology is very well explained, allowing any researcher to reproduce the study if they follow the described steps. The methodology is suitable for achieving the proposed objectives. The statistics used are appropriate. However, a flowchart of the study participants is missing.

The results are well described and supported by tables and figures that make them easier to read and understand, and they align with the stated objectives. The discussion is well-developed but is based on a limited number of references, only seven, which weakens the results. We recommend expanding it both in text and in bibliographic references. In the last paragraph, conclusions are discussed, but we believe it would be more appropriate to have them in a separate section.

The bibliographic references, while appropriate, are somewhat outdated:

  • The overall obsolescence index (median age of the references) is 15 years, which is considered very high and should be reduced.
  • The obsolescence index of the introduction is also high (12 years), while that of the discussion is only 4 years (which is adequate but should be expanded). No self-citations are noted.

Author Response

Reviewer 2

The article addresses the association of topics of significant interest to the journal's readers, such as physical performance, quality of life, disability, and low back pain in obese individuals. The article is well-structured and contains all the sections of an original research paper.

The introduction is well-crafted, although it is somewhat brief, which does not facilitate a clear focus on the problem. The methodology is very well explained, allowing any researcher to reproduce the study if they follow the described steps. The methodology is suitable for achieving the proposed objectives. The statistics used are appropriate.

We acknowledge the Reviewer for carefully evaluating our manuscript, which is aimed at improving the quality of our work further.

Q1. However, a flowchart of the study participants is missing.

As requested, a flow chart of the study participants has been added.

Lines (changes made): 98-102

Q2. The results are well described and supported by tables and figures that make them easier to read and understand, and they align with the stated objectives. The discussion is well-developed but is based on a limited number of references, only seven, which weakens the results. We recommend expanding it both in text and in bibliographic references. In the last paragraph, conclusions are discussed, but we believe it would be more appropriate to have them in a separate section.

A2. As recommended, the number of references for the discussion have been increased. As suggested, conclusions have been moved to a separate section.

Lines (changes made): 288-290, 301-304

Q3. The bibliographic references, while appropriate, are somewhat outdated:

The overall obsolescence index (median age of the references) is 15 years, which is considered very high and should be reduced.

The obsolescence index of the introduction is also high (12 years), while that of the discussion is only 4 years (which is adequate but should be expanded). No self-citations are noted.

A3. As recommended, references have been updated.

Lines (changes made): 36, 42, 45, 47

Round 2

Reviewer 1 Report

Comments and Suggestions for Authors

*

Author Response

Thank you for reviewing our paper and for your valuable comments, which significantly improved our paper. 

Reviewer 2 Report

Comments and Suggestions for Authors

Is ok

Author Response

(The authors gave the same response as above.)
